# LUMINA☀: DETECTING HALLUCINATIONS IN RAG SYSTEM WITH CONTEXT–KNOWLEDGE SIGNALS

**Samuel Yeh[1,2], Sharon Li[1], Tanwi Mallick[2]**
[1]Department of Computer Science, University of Wisconsin-Madison
{samuelyeh, sharonli}@cs.wisc.edu
[2]Argonne National Laboratory
tmallick@anl.gov

## ABSTRACT

Retrieval-Augmented Generation (RAG) aims to mitigate hallucinations in large language models (LLMs) by grounding responses in retrieved documents. Yet, RAG-based LLMs still hallucinate even when provided with correct and sufficient context. A growing line of work suggests that this stems from an imbalance between how models use external context and their internal knowledge, and several approaches have attempted to quantify these signals for hallucination detection. However, existing methods require extensive hyperparameter tuning, limiting their generalizability. We propose LUMINA, a novel framework that detects hallucinations in RAG systems through *context–knowledge signals*: external context utilization is quantified via distributional distance, while internal knowledge utilization is measured by tracking how predicted tokens evolve across transformer layers. We further introduce a framework for statistically validating these measurements. Experiments on common RAG hallucination benchmarks and four open-source LLMs show that LUMINA achieves consistently high AUROC and AUPRC scores, outperforming prior utilization-based methods by up to +13% AUROC on HalluRAG. Moreover, LUMINA remains robust under relaxed assumptions about retrieval quality and model matching, offering both effectiveness and practicality.

**LUMINA:** **https://github.com/deeplearning-wisc/LUMINA**

## 1 INTRODUCTION

Large language models (LLMs) are prone to hallucination, *i.e.*, producing responses that are factually incorrect, nonsensical, or not grounded in the input or available data, while still appearing fluent and plausible (Luo et al., 2024; Huang et al., 2024; Park et al., 2025). One commonly used strategy to mitigate hallucination is providing LLMs with relevant information retrieved from external knowledge bases, so-called Retrieval-Augmented Generation (RAG) (Shuster et al., 2021; Fan et al., 2024; Gao et al., 2024). However, despite having sufficient and relevant retrieved documents, RAG systems still have a chance to hallucinate and produce statements that are either unsupported or contradict the retrieved information (Niu et al., 2024; Ridder & Schilling, 2025).

Recent work has shown that such failures often arise from conflicts between an LLM's internal knowledge and the retrieved external context (Xu et al., 2024). In these cases, models tend to over-rely on internal knowledge regardless of correctness, undermining factual reliability (Longpre et al., 2021; Li et al., 2023; Sun et al., 2025a; Yamin et al., 2025). Inspired by this observation, recent approaches attempt to quantify hallucinations in RAG (Sun et al., 2025b; Wang, 2025; Tao et al., 2025). However, existing methods rely on mechanistic interpretability heuristics—such as selecting specific attention heads or transformer layers to achieve the optimal hallucination detection performance—which require heavy hyperparameter tuning and often fail to generalize across models and datasets.

To overcome these limitations, we propose LUMINA, a new framework for detecting hallucinations in RAG system through *context–knowledge signals*, namely the signals of external context utilization and internal knowledge utilization, as shown in Figure 1. Rather than targeting particular attention heads

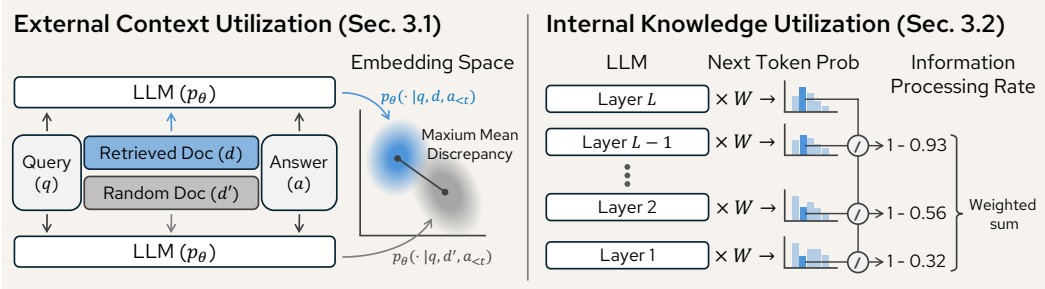

Figure 1: **The overview of LUMINA.** For external context utilization, we propose to measure the maximum mean discrepancy between two next token probability distributions conditioned on different documents. For internal knowledge utilization, we introduce the idea of information processing rate by looking at the ratio of the most probable output token's probability across transformer layers and use it to determine the amount of utilized internal knowledge when generating the next token.

or layers, LUMINA measures these signals in a layer-agnostic manner, requiring less hyperparameter tuning. Specifically, for **external context utilization**, we measure the discrepancy between predictive distributions conditioned on retrieved documents *vs.* random documents. A larger discrepancy indicates that the LLM is more sensitive to semantic changes in documents when generating the answer, implying higher reliance on the external context. For **internal knowledge utilization**, we track how the model's internal states and token predictions evolve across layers: if the internal layers' predictions do not converge to the final output until later layers, it suggests more information is added during the layer-wise process, implying stronger reliance on internal knowledge. We further validate the soundness of our measurements through statistical hypothesis testing on verifiable implications, establishing a stronger link between the proposed scores and actual utilization.

We conduct extensive experiments on common RAG hallucination benchmarks and across four LLMs to evaluate the performance of LUMINA on hallucination detection. The results show that the hallucination score calculated with LUMINA outperforms existing methods by a significant margin. For example, LUMINA achieves more than 0.9 AUROC on the HalluRAG datasets across models, with improvements of up to **+13**% over prior state-of-the-art. Importantly, the decomposition into external context utilization and internal knowledge utilization provides interpretable insights: hallucinations are strongly associated with low external context scores and disproportionately high internal knowledge scores. We further demonstrate that LUMINA is robust across different retrieval settings. These results validate both the effectiveness and practicality of our framework.

Our key contributions are summarized as follows:

1. We propose LUMINA, a novel approach to quantify utilization of external context and internal knowledge for RAG-based hallucination detection.
2. We propose a framework to statistically validate LUMINA, showing that they align with the intended results.
3. We conduct extensive experiments and show that LUMINA outperforms both score-based and learning based methods in hallucination detection, establishing new *state-of-the-art*.

## 2 PRELIMINARIES

### 2.1 PROBLEM FORMULATION AND MOTIVATION

RAG systems aim to improve factuality by incorporating external documents into the generation process. In cases such as news summarization, information extraction given a json file, and question answering that requires information emerging after the model's release date, RAG is usually necessary because an LLM cannot rely solely on its internal knowledge to complete the task. However, in such cases, hallucinations still occur when a model over-relies on its internal parametric knowledge and under-utilizes the retrieved external context. We provide a formal definition below.

**Conjecture 1** (**External context vs. internal knowledge utilization**). *Let $p_\theta$ be an RAG-based LLM that takes a query $q$ and retrieved documents $d$ as inputs to generate a response $a$. Assume $d$ is relevant to $q$ and contains correct and sufficient information to respond to $q$. Denote $\mathcal{E}_{p_\theta}(a|q,d), \mathcal{I}_{p_\theta}(a|q,d) \in \mathbb{R}$ be the signals of external context utilization and internal knowledge utilization of $p_\theta$, respectively, when generating $a$. The response $a$ is more likely to be hallucination if $\mathcal{I}_{p_\theta}(a|q,d) \gg \mathcal{E}_{p_\theta}(a|q,d)$.*

Conjecture 1 is built on a principled intuition that, if a LLM requires external knowledge to complete a task and if a retriever can provide the LLM sufficient external information, the LLM should utilize those external context and ground its reasoning ability on those context. Therefore, a response in this scenario will be considered less reliable if it disproportionally relies on the LLM's internal knowledge without a sufficient amount of external knowledge utilization.

**Definition 2.1** (**Hallucination in an RAG system**). *Based on Conjecture 1, we define hallucination scores at both the token and response level. Specifically, for a generated answer $a = (a_1, \ldots, a_T)$ with $T$ tokens, let $\mathcal{E}_{p_\theta}(a_t|q,d,a_{<t}), \mathcal{I}_{p_\theta}(a_t|q,d,a_{<t}) \in \mathbb{R}$ be the signals of external context utilization and internal knowledge utilization of $p_\theta$ when generating the token $a_t$, respectively. The token-level hallucination score of $a_t$ is defined as*

$$\mathcal{H}_t(a_t|q,d,a_{<t}) := \lambda \cdot \mathcal{I}_{p_\theta}(a_t|q,d,a_{<t}) - (1-\lambda) \cdot \mathcal{E}_{p_\theta}(a_t|q,d,a_{<t}), \tag{1}$$

*where $\lambda$ is a hyperparameter. Similarly, the response-level hallucination score of the response $a$ is defined as the average of the token-level hallucination scores, i.e.,*

$$\mathcal{H}_r(a|q,d) := \frac{1}{T}\sum_{t=1}^{T}\mathcal{H}_t(a_t|q,d,a_{<t}). \tag{2}$$

In this paper, we focus on the core question: ***How to quantify the utilization of external context and internal knowledge?***

## 2.2 RELATED WORK

Prior works have attempted to quantify $\mathcal{E}_{p_\theta}(a_t|q,d,a_{<t})$ and $\mathcal{I}_{p_\theta}(a_t|q,d,a_{<t})$ using empirical metrics (Sun et al., 2025b; Wang, 2025). For example, Sun et al. (2025b) proposed ReDeEP, which measures external context utilization through cosine similarity between the generated token and tokens in context that have high attention weights w.r.t. certain attention heads. For internal knowledge utilization, it measures the Jensen-Shannon (JS) divergence between the hidden states before/after the FFN layer of certain transformer layers. The success of ReDeEP on some RAG hallucination detection datasets validates the idea of Conjecture 1. Wang (2025) combine the idea of ReDeEP with semantic entropy probes (SEP) (Han et al., 2024). They quantified external context utilization by measuring the semantic correlation between the semantic entropy of the generated token and attended tokens in the context. For internal knowledge utilization, they measured the absolute difference between the semantic entropy corresponding to hidden states before and after the FFN layer.

Although these approaches effectively detect hallucinations in the RAG system, they have two major limitations. First, these approaches require selecting specific attention heads and transformer layers to compute the external context score and internal knowledge score. However, the selection process is non-trivial and requires extensive hyperparameter tuning. In addition, these hyperparameters are dataset and model-specific, limiting the generalizability across different datasets and models. Another limitation is that although these works demonstrated the correlation between their proposed scores and hallucination, they did not validate whether the scores truly reflect the utilization of external context and internal knowledge.

## 3 METHODOLOGY

**Overview.** To overcome the limitations of prior empirical approaches, we introduce LUMINA, a new framework for quantifying both external context and internal knowledge utilization. In Section 3.1 and Section 3.2, we formalize the quantification of the two signals, which will be combined to compute the final hallucination score. In Section 3.3, we propose to validate the soundness of LUMINA through extensive hypothesis testing, addressing the challenges of score validation in previous works.

### 3.1 QUANTIFYING EXTERNAL CONTEXT UTILIZATION

To measure LLM's external context utilization, our key idea is to assess its sensitivity to semantic changes in the input documents. If the LLM effectively incorporates the external context to generate a response, then replacing relevant documents with random ones should noticeably change the token probability distribution. Formally, we propose the following measurement:

**Measurement 1** (**External context utilization**)**.** *Let $a$ be an LLM-generated answer to query $q$ with retrieved documents $d$ as input. Assume $d$ is relevant to $q$ and contains correct and sufficient information to respond to $q$. Let $d'$ be a subset of random documents irrelevant to $q$. The model's predictive distribution over tokens induces two (approximated) distributions over embeddings:*

$$P(E_v) = p_\theta(v \mid q, d, a_{<t}), \quad Q(E_v) = p_\theta(v \mid q, d', a_{<t}), \tag{3}$$

*where each token $v \in \mathcal{V}$ in the vocabulary space is associated with an embedding $E_v \in \mathbb{R}^D$. Then, the degree to which the model uses external context for generating token $a_t$ is reflected in the divergence between the two distributions conditioned on $d$ versus $d'$:*

$$\mathcal{E}_{p_\theta}(a_t | q, d, a_{<t}) := \Delta(P, Q), \tag{4}$$

*where $\Delta : \mathcal{P} \times \mathcal{P} \to \mathbb{R}_+$ is a distance function between two probability distributions.*

Note that we adopt $P(E_v)$ and $Q(E_v)$ as proxies to approximate the ground truth embedding distribution, as it is challenging to estimate it over the high-dimensional vector space. We instantiate $\Delta$ with Maximum Mean Discrepancy (MMD), which measures the distance of two probability distributions by mapping them into a Reproducing Kernel Hilbert Space.

**Definition 3.1** (**Maximum Mean Discrepancy** (Gretton et al., 2012))**.** *Given a positive semi-definite kernel function $k$, the squared MMD between two probability distributions $P$ and $Q$ is defined as*

$$\mathrm{MMD}_k^2(P, Q) := \mathbb{E}_{\boldsymbol{A}, \boldsymbol{A}' \sim P}[k(\boldsymbol{A}, \boldsymbol{A}')] + \mathbb{E}_{\boldsymbol{B}, \boldsymbol{B}' \sim Q}[k(\boldsymbol{B}, \boldsymbol{B}')] - 2\mathbb{E}_{\boldsymbol{A} \sim P, \boldsymbol{B} \sim Q}[k(\boldsymbol{A}, \boldsymbol{B})], \tag{5}$$

*where $\boldsymbol{A}, \boldsymbol{A}'$ are i.i.d. vectors randomly sampled from $P$ and $\boldsymbol{B}, \boldsymbol{B}'$ are sampled from $Q$.*

This metric provides us with a non-parametric and LLM-agnostic way to quantify the utilization of external context, making it generalizable to different models and datasets.

By rewriting MMD with $P$ and $Q$ we defined in Eq. (3) over token embeddings, we obtain:

$$\begin{aligned}
\mathcal{E}_{p_\theta}(a_t | q, d, a_{<t}) := &\sum_{u,v \in \mathcal{V}} P(E_u)P(E_v)k(E_u, E_v) + \sum_{u,v \in \mathcal{V}} Q(E_u)Q(E_v)k(E_u, E_v) \\
&- 2\sum_{u,v \in \mathcal{V}} P(E_u)Q(E_v)k(E_u, E_v).
\end{aligned} \tag{6}$$

We adopt the cosine kernel:

$$k_{\cos}(E_u, E_v) := \frac{1}{2}\left(1 + \frac{E_u^T E_v}{\|E_u\|_2 \|E_v\|_2}\right). \tag{7}$$

Note that the cosine kernel acts equivalent to computing cosine similarity between two token embeddings, which is commonly used to measure the semantic similarity of two pieces of text. In Section 4.4, we experiment with alternative kernels such as the Gaussian kernel, and we show that our method is not sensitive to the choice of kernels.

### 3.2 QUANTIFYING INTERNAL KNOWLEDGE UTILIZATION

To quantify the utilization of internal knowledge, we focus on the signals in internal states of an LLM. Specifically, a transformer-based autoregressive LLM has multiple layers, through which information is gradually added into a residual stream that flows from the input layer to the output layer, shaping the output token representation and probability distribution (Geva et al., 2022). Studies have found that by projecting the hidden state of each layer to the token representation space, we can interpret what an LLM believes after the process of each layer (nostalgebraist, 2020). In addition, via logit lens (nostalgebraist, 2020), studies have identified the saturation event in an LLM, *i.e.*, the top-$k$

prediction of the LLM remains constant in all subsequent layers after a certain layer called the $k$-th saturation layer (Geva et al., 2022; Lioubashevski et al., 2025).

Inspired by these observations, we propose a metric that quantifies how actively the model updates its predictions across layers. Formally, we define the rate of information processing below.

**Definition 3.2** (**Information processing rate**). *Given an LLM $p_\theta$ with $L$ layers, which takes $x_{<t}$ as the input and generate the next token $x_t$, we denote $x_{t,1} := \arg\max_v p_\theta(v|x_{<t})$ as the most probable next token and $h_{t,l} \in \mathbb{R}^D$ as the $l$-th layer hidden state when generating $x_t$. Let $f : \mathbb{R}^D \to \mathcal{P}$ be a projection from a hidden state to a probability distribution over the vocabulary $\mathcal{V}$. The information processing rate of $p_\theta$ conditioned on $x_{<t}$ is defined as*

$$\mathcal{R}_{p_\theta}(x_{<t}) := \frac{\sum_{l=1}^{L-1} \left(1 - \min\left\{\frac{[f(h_{t,l})]_{x_{t,1}}}{p_\theta(x_{t,1}|x_{<t})}, 1\right\}\right) \cdot l}{\sum_{l'=1}^{L-1} \frac{l'}{H(f(h_{t,l'}))}}, \tag{8}$$

where $H(\cdot)$ is the entropy function, and $f$ is the logit lens (nostalgebraist, 2020) that projects the hidden state of each layer to logits using the LayerNorm and the unembedding matrix $\boldsymbol{W}$, *i.e.*,

$$\mathrm{LogitLens}(h) := \mathrm{LayerNorm}(h)\boldsymbol{W}, \quad f(\cdot) := \mathrm{Softmax}(\mathrm{LogitLens}(\cdot)). \tag{9}$$

Specifically, $\mathcal{R}_{p_\theta}(x_{<t})$ captures two key elements: (1) The numerator measures the extent to which each layer's prediction for the most probable token differs from the final output, weighted by layer depth to emphasize later-layer processing. When $\frac{[f(h_{t,l})]_{x_{t,1}}}{p_\theta(x_{t,1}|x_{<t})}$ is small, it indicates the layer has not yet converged to the final prediction, suggesting active information processing. (2) The denominator provides adaptive normalization based on each layer's prediction uncertainty (entropy), giving higher relative weight to layers that exhibit confident, decisive processing patterns. Given this definition, we attribute the utilization of internal knowledge to the 1st information processing rate and propose the following measurement:

**Measurement 2** (**Internal knowledge utilization**). *An LLM is considered to be more heavily utilizing its internal knowledge to generate $a_t$ when it exhibits a higher information processing rate. Specifically, we propose that the internal knowledge utilization of an LLM to generate $a_t$ given $q$ and $d$ can be measured as*

$$\mathcal{I}_{p_\theta}(a_t|q, d, a_{<t}) := \mathcal{R}_{p_\theta}(q, d, a_{<t}). \tag{10}$$

### 3.3 Statistical Validation of the Measurement

In this section, we validate the soundness of our approach. Previous work such as Sun et al. (2025b) primarily verified whether their scores have a causal relationship with hallucination but failed to show the relationship between the scores and actual external context/internal knowledge utilization. To address this, we directly assess whether our measurements capture the intended notion of utilization. Specifically, we derive verifiable implications that must hold if our proposed measurements are valid. We then use the proposed score to verify these implications with statistical hypothesis testing. If the proposed score passes all tests, the score reflects the corresponding utilization.

**External context utilization.** To validate Measurement 1, we examine the following implications:

**H1.** If Measurement 1 is valid, then $\mathcal{E}_{p_\theta}(a_t|q, d, a_{<t}) > \mathcal{E}_{p_\theta}(a'_t|q, \varnothing, a'_{<t})$. That is, generations with retrieved documents have stronger external context utilization than generations without.

**H2.** If Measurement 1 is valid, then $\mathcal{E}_{p_\theta}(a_t|q_{\mathrm{sum}}, d_{\mathrm{sum}}, a_{<t}) > \mathcal{E}_{p_\theta}(a_t|q_{\mathrm{QA}}, d_{\mathrm{QA}}, a_{<t})$. That is, summarization tasks should exhibit higher external context utilization than question answering.

**Internal knowledge utilization.** To validate Measurement 2, we examine the following:

**H3.** If Measurement 2 is valid, then $\mathcal{R}^1_{p_\theta}(q, \varnothing, a_{<t}) > \mathcal{R}^1_{p_\theta}(q, d, a_{<t})$. That is, generating an answer without retrieved documents requires more internal knowledge than with retrieved documents.

**H4.** If Measurement 2 is valid, then $\mathcal{R}^1_{p_\theta}(q_{\mathrm{D2T}}, d_{\mathrm{D2T}}, a_{<t}) > \mathcal{R}^1_{p_\theta}(q_{\mathrm{sum}}, d_{\mathrm{sum}}, a_{<t})$. In other words, data-to-text generation requires more internal knowledge than summarization.

To examine **H1**, we utilize data in the QA set of RAGTruth (Niu et al., 2024). We use the original data to compute $\mathcal{E}_{p_\theta}(a_t|q, d, a_{<t})$, and generate additional answers without providing retrieved documents

Table 1: **All the hypotheses pass the statistical tests.** For H1, H2, H4, we report one-tailed t-statistic; for H3, we report paired-sample one-tailed t-statistic. All four implications reject their null hypothesis, validating the soundness of LUMINA. Note that the tests are run with $> 65k$ tokens and the magnitude of the t-statistic means how easy we can distinguish the two distributions. * $p < 0.05$; ** $p < 0.01$; *** $p < 0.001$.

| LLM | H1 | H2 | H3 | H4 |
|---|---|---|---|---|
| Llama2-7B | 79.85*** | 27.67*** | 101.20*** | 15.36*** |
| Llama2-13B | 73.49*** | 20.51*** | 91.00*** | 7.71*** |
| Llama3-8B | 94.15*** | 6.35*** | 102.44*** | 15.85*** |
| Mistral-7B | 88.70*** | 6.21*** | 109.26*** | 9.69*** |

as $a'$ to compute $\mathcal{E}_{p_\theta}(a'_t|q, \varnothing, a'_{<t})$. For **H2**, we utilize the Summary and QA set of RAGTruth; for **H4**, the Summary and Data2Text set; and for **H3**, the entire RAGTruth dataset. We test the hypotheses with four different instruction-tuned LLMs, including Llama2-{7B, 13B} (Llama Team, 2023), Llama3-8B (Llama Team, 2024), and Mistral-7B (Jiang et al., 2023). Results in Table 1 indicate that all four implications reject their null hypothesis, validating our measurements for external context utilization and internal knowledge utilization.

## 4 EXPERIMENTS

### 4.1 EXPERIMENTAL SETTINGS

**Baselines.** We compare LUMINA with baselines across 8 different hallucination detection strategies: (1) **Uncertainty-based**, which detects hallucination by estimating uncertainty via token-level probability or entropy. Baselines of this category include Perplexity (Ren et al., 2023), LN-Entropy (Malinin & Gales, 2021), and Focus (Zhang et al., 2023). (2) **Cross-sample consistency**, which detects hallucination by sampling multiple responses for a query and measuring their (logic/semantic) consistency. Approaches include SelfCKGPT (Manakul et al., 2023) and EigenScore (Chen et al., 2024). (3) **Verbalization**, which detects hallucinations by prompting another LLM to score the correctness of the answer. Approaches include P(True) (Kadavath et al., 2022) and RefChecker (Hu et al., 2024). (4) **Utilization of external context and internal knowledge**, which decouples these two signals via findings in the study of mechanistic interpretability. Baseline of this category is ReDeEP (Sun et al., 2025b). Details of each baseline are introduced in Appendix B.

**LLMs.** To demonstrate the generalizability of LUMINA, we conduct experiments with four open-sourced LLMs, including Llama2-{7B, 13B}, Llama3-8B, and Mistral-7B. Specifically, each LLM is used to detect hallucinations in responses generated by the same model. We also report the performance of proxy LLM setting, *i.e.*, using one LLM to detect hallucinations in responses generated by another model, in Sec. 4.3. All LLMs are the instruction-tuned version.

**Datasets.** Experiments are conducted on two representative RAG hallucination detection benchmarks: **RAGTruth** (Niu et al., 2024), the first high-quality RAG hallucination detection dataset, consisting of three types of RAG tasks, including question answering, data-to-text writing, and news summarization. **HalluRAG** (Ridder & Schilling, 2025), a dataset of free-form question answering in an RAG setting. Details of these datasets are introduced in Appendix C.

**Evaluation metrics.** We measure the performance with three metrics: **AUROC**, **AUPRC**, and **Pearson's correlation coefficient** (PCC). AUPRC captures precision-recall trade-offs, while AUROC evaluates the trade-offs between true and false positive rates. These metrics are threshold-agnostic and better suited for comparing scoring-based methods. We also report the optimal precision, recall, and F1 score ($\text{Prec}_{\text{Opt}}, \text{Recall}_{\text{Opt}}, \text{F1}_{\text{Opt}}$) in Appendix E.1, where $\text{F1}_{\text{Opt}}$ is the optimal F1 score among all possible threshold and $\text{Prec}_{\text{Opt}}$ and $\text{Recall}_{\text{Opt}}$ are corresponding Precision and Recall.

**Implementation details.** We adopt $\lambda = 0.5$ to compute Eq. (1) as ablations show that balancing the scores of external context and internal knowledge yields relatively strong performance (see Appendix E.3 for detailed ablations). Other implementation details and computational resources of LUMINA are reported in Appendix D and G, respectively.

Table 2: **LUMINA consistently achieves a high performance across datasets and LLMs.** The highest scores are set in **bold**. Note that HalluRAG dataset does not contain responses generated by Llama3-8B.

| LLM | Approach | RAGTruth | | | HalluRAG | | |
|---|---|---|---|---|---|---|---|
| | | AUROC ↑ | PCC ↑ | AUPRC ↑ | AUROC ↑ | PCC ↑ | AUPRC ↑ |
| Llama2-7B | Perplexity | 0.5103 | -0.0118 | 0.4836 | 0.4610 | -0.0673 | 0.2332 |
| | LN-Entropy | 0.6964 | 0.3318 | 0.6615 | 0.9102 | 0.5133 | 0.6812 |
| | Focus | 0.5633 | 0.0811 | 0.5386 | 0.5652 | 0.2415 | 0.3844 |
| | SelfCKGPT | 0.4787 | -0.0279 | 0.4859 | 0.4669 | -0.0070 | 0.2377 |
| | EigenScore | 0.5454 | 0.0717 | 0.5183 | 0.6720 | 0.2705 | 0.4470 |
| | P(True) | 0.5197 | 0.0404 | 0.5334 | 0.5847 | 0.1143 | 0.2976 |
| | RefChecker | 0.5869 | 0.1751 | 0.6827 | 0.4907 | -0.0255 | 0.2750 |
| | ReDeEP | 0.7273 | 0.3859 | 0.6971 | 0.6771 | 0.1468 | 0.3378 |
| | LUMINA | **0.7646** | **0.4546** | **0.7491** | **0.9153** | **0.6554** | **0.7572** |
| Llama2-13B | Perplexity | 0.4539 | -0.1020 | 0.3993 | 0.2548 | -0.2366 | 0.0944 |
| | LN-Entropy | 0.7677 | 0.4446 | 0.6838 | 0.7826 | 0.3262 | 0.3567 |
| | Focus | 0.5451 | 0.0130 | 0.4603 | 0.6739 | 0.2563 | 0.3181 |
| | SelfCKGPT | 0.4545 | -0.0835 | 0.4106 | 0.7729 | 0.2640 | 0.3029 |
| | EigenScore | 0.6329 | 0.2080 | 0.5202 | 0.7862 | 0.4250 | 0.4867 |
| | P(True) | 0.7543 | 0.3821 | 0.7418 | 0.6914 | 0.2480 | 0.2146 |
| | RefChecker | 0.6363 | 0.2723 | 0.6988 | 0.5670 | 0.1390 | 0.3169 |
| | ReDeEP | 0.8055 | 0.5195 | 0.7792 | 0.7645 | 0.2705 | 0.3001 |
| | LUMINA | **0.8569** | **0.6041** | **0.8436** | **0.9166** | **0.6044** | **0.8497** |
| Llama3-8B | Perplexity | 0.7130 | 0.3568 | 0.7183 | - | - | - |
| | LN-Entropy | 0.7072 | 0.3500 | 0.7109 | - | - | - |
| | Focus | 0.5258 | 0.0375 | 0.5380 | - | - | - |
| | SelfCKGPT | 0.5339 | 0.0491 | 0.5550 | - | - | - |
| | EigenScore | 0.6001 | 0.1774 | 0.5824 | - | - | - |
| | P(True) | 0.5407 | 0.0928 | 0.5502 | - | - | - |
| | RefChecker | 0.5718 | 0.1494 | 0.6874 | - | - | - |
| | ReDeEP | **0.7495** | **0.4458** | 0.7817 | - | - | - |
| | LUMINA | 0.7446 | 0.4236 | **0.7874** | - | - | - |
| Mistral-7B | Perplexity | 0.6200 | 0.1463 | 0.6106 | 0.5362 | -0.0264 | 0.1261 |
| | LN-Entropy | 0.7607 | 0.4386 | 0.7377 | 0.9188 | 0.6076 | 0.7347 |
| | Focus | **0.7803** | 0.4188 | 0.7647 | 0.8565 | 0.4318 | 0.4219 |
| | SelfCKGPT | 0.5680 | 0.0812 | 0.5698 | 0.8275 | 0.5552 | 0.6098 |
| | EigenScore | 0.5642 | 0.1006 | 0.5637 | 0.8652 | 0.6411 | 0.7337 |
| | P(True) | 0.7530 | 0.4334 | 0.7494 | 0.5899 | 0.0886 | 0.1771 |
| | RefChecker | 0.6017 | 0.2047 | 0.7303 | 0.5065 | 0.0153 | 0.1784 |
| | ReDeEP | 0.7615 | 0.4613 | **0.8133** | 0.7870 | 0.2611 | 0.3516 |
| | LUMINA | 0.7685 | **0.4623** | 0.7942 | **0.9899** | **0.7529** | **0.9431** |

## 4.2 MAIN RESULTS

**LUMINA achieves state-of-the-art performance.** Table 2 summarizes the experimental comparison across methods. The results show that LUMINA has a consistently high performance across datasets and LLMs. In particular, it almost always outperforms ReDeEP, the previous attempt of measuring the utilization of external context and internal knowledge to detect hallucinations. The gap between them is particularly large on the HalluRAG dataset. Noticeably, LUMINA achieves more than 0.9 AUROC on the HalluRAG dataset across models, outperforming the baselines by a substantial margin. We further conduct an error analysis to see when and why LUMINA fails. Specifically, we sample 20 false-negative and 20 false-positive cases from the RAGTruth dataset, respectively, and qualitatively analyze the reason of errors. The result reveals that most of the errors stem from incorrect labels and low-quality retrieved documents of the dataset, suggesting a potentially higher performance in a setting with high-quality data. The details of this analysis can be found in Appendix F.

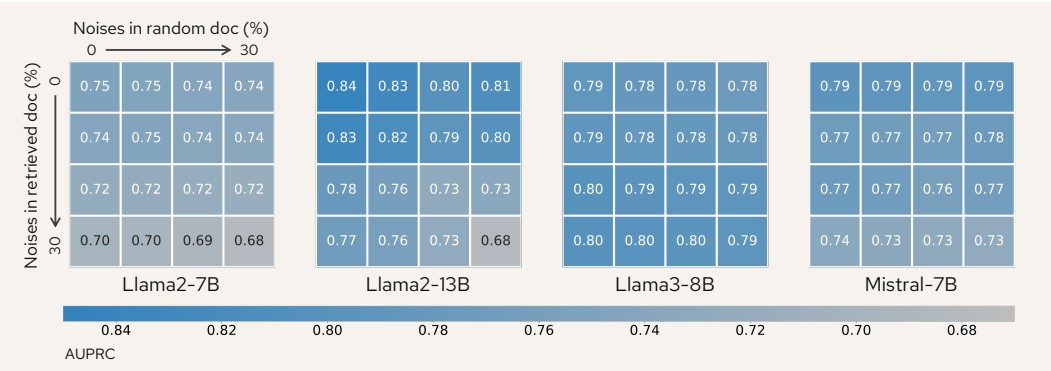

Figure 2: **Noises in context do not largely degrade the performance of LUMINA.** We add $0 \sim 30\%$ noises to the retrieved documents and random documents and evaluate the hallucination detection performance. The experiment is conducted on the RAGTruth dataset.

**Comparison with supervised approach.** We also compare LUMINA with SAPLMA (Azaria & Mitchell, 2023), a supervised approach that trained a binary classifier on the last token hidden states to detect hallucination. Since our method is unsupervised in nature and does not rely on labeled data, the supervised baseline can be viewed as a performance upper bound. Results in Appendix E.2 show that LUMINA achieves a competitive performance against SAPLMA and even sometimes outperforms it, all without any training, highlighting both its supreme performance and ease of deployment.

## 4.3 RELAXING ASSUMPTIONS

In Section 3, we implicitly make two assumptions: 1) *perfect context assumption:* we assume the retrieved documents $d$ are correct, sufficient, and relevant to the query. 2) *same LLM assumption:* we assume the LLM used to compute the external context score and internal knowledge score is the same as the LLM used to generate responses. These two assumptions are usually introduced in other hallucination detection works as well (Zhang et al., 2023; Sun et al., 2025b). Unfortunately, they are often strong and have a significant impact on the performance, limiting the usability of these methods (such as for open-sourced model-generated responses only). In this section, we investigate the performance of LUMINA when relaxing these two assumptions, showing the robustness of LUMINA.

**Relaxing perfect context assumption.** We relax this assumption by gradually injecting noise into the retrieved documents $d$ and random documents $d'$. Specifically, for the assumption on retrieved documents, we randomly remove $\{0\%, 10\%, 20\%, 30\%\}$ sentences from $d$. And for the assumption on the random documents, we randomly add $\{0\%, 10\%, 20\%, 30\%\}$ sentences from $d$ to $d'$. Figure 2 shows the AUPRC of all noise injection combinations on the RAGTruth dataset. The result shows that except Llama2-13B, which has a $> 0.1$ performance drop after injecting noises, LUMINA with other LLMs yields stable performance. Furthermore, after removing sentences from retrieved documents, LUMINA with Llama3-8B even achieves a higher AUPRC. These results demonstrate the robustness of LUMINA against context noises.

**Relaxing the same LLM assumption.** We relax this assumption by using different LLMs to compute the scores for a response. Specifically, we use Llama2-7B, Llama2-13B, Llama3-8B, Mistral-7B, Qwen3-4B (Yang et al., 2025), Llama4-17B (MetaAI, 2025) to detect hallucination on the RAGTruth dataset, which contains responses generated by Llama2-7B, Llama2-13B, Llama2-70B, Llama3-8B, Mistral-7B, GPT-3.5, and GPT-4. Figure 3 shows AUROC across different generator-detector LLM pairs.

The results show that the same model setting is not always necessary. Specifically, Llama2-7B achieves a comparable or higher AUROC than Llama3-8B on Llama3-8B responses. Moreover, LUMINA with Llama2-7B and Llama3-8B has stable performance across different generation LLMs. In addition, newer models, such as Qwen3-4B and Llama4-17B, also perform well across generation LLMs. Overall, LUMINA demonstrates a plausible solution for generation LLM-agnostic hallucination detection, which is more practical in real-world scenarios.

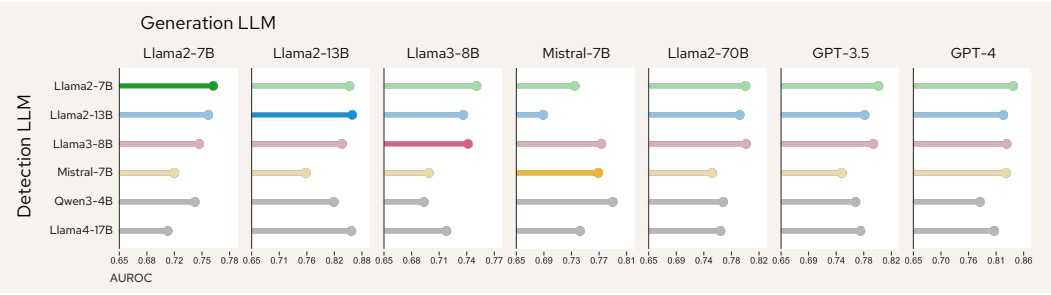

Figure 3: **The "same LLM" setting is not essential for LUMINA to achieve the optimal performance.** On the RAGTruth dataset, for each set of responses generated by the same LLM, we apply LUMINA with a different base LLM to detect hallucination. Bars in more saturated shades indicate settings where the same LLM is used for both generation and detection.

## 4.4 ABLATION STUDY

**Impact of kernel selection.** We ablate on the selection of kernel $k \in \{\text{Cosine}, \text{RBF}_{0.5}, \text{RBF}_{0.7}, \text{RBF}_1, \text{RBF}_2, \text{RBF}_3\}$, where $\text{RBF}_\sigma$ is a RBF kernel, *i.e.*, $\text{RBF}_\sigma(E_u, E_v) := \exp\left(-\frac{\|E_u - E_v\|_2^2}{2\sigma^2}\right)$. Figure 4 shows the AUPRC of different kernels on the RAGTruth dataset. The results show that the optimal setting of the RBF kernel has a similar performance to the cosine kernel, suggesting our external context score is insensitive to the kernel selection. We default to the cosine kernel as it is less dependent on hyperparameters, making it easy to use in practice.

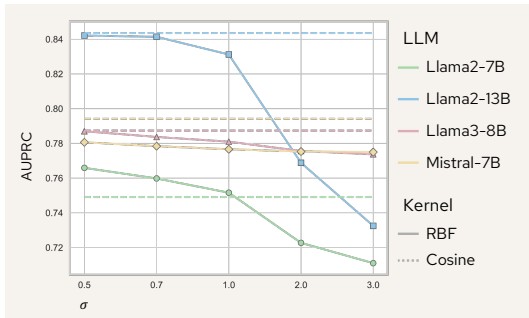

Figure 4: **MMD with cosine kernel performs similarly or better than with RBF kernel.**

**Impact of external context & internal knowledge.** Our final hallucination score is the combination of the external context score and internal knowledge score. To obtain more insights into how each component contributes to the final score, we ablate on the components by considering only the external context score and internal knowledge score. The right plot of Figure 5 shows that combining scores of external context and internal knowledge achieves the highest AUPRC on the RAGTruth dataset for every LLM. For example, on Llama2-13B, the combination leads to more than 10% improvement. This observation justifies the effectiveness of the hallucination score introduced in Definition 2.1. In addition, the left plot of Figure 5 shows that a response generated by Llama2-13B is more likely to be hallucination if it has a high internal knowledge score and a low external context score. This observation validates Conjecture 1 and suggests that Eq. (1) does not imply an objective function that forces LLM only using external context to answer questions. Instead, it suggests that the internal knowledge utilization should be grounded in an external context to achieve a reliable generation, implying its potential for generalizing to reasoning-intensive tasks.

**Additional ablations.** We also conduct other ablations, covering the selection of $\lambda$ in Eq (1), the impact of random documents $d'$, and the contribution of two components of the information processing rate. Please see Appendix E.3 and E.4 for more details.

## 5 CONCLUSION

In this paper, we introduce LUMINA, a novel approach to quantify the utilization of external context and internal knowledge. These context–knowledge signals provide a principled way to assess how LLMs balance retrieved evidence against their own parametric knowledge during generation. Experimental results on common benchmarks across four LLMs demonstrate that LUMINA has a consistently high performance on hallucination detection for RAG-based generations, outperforming

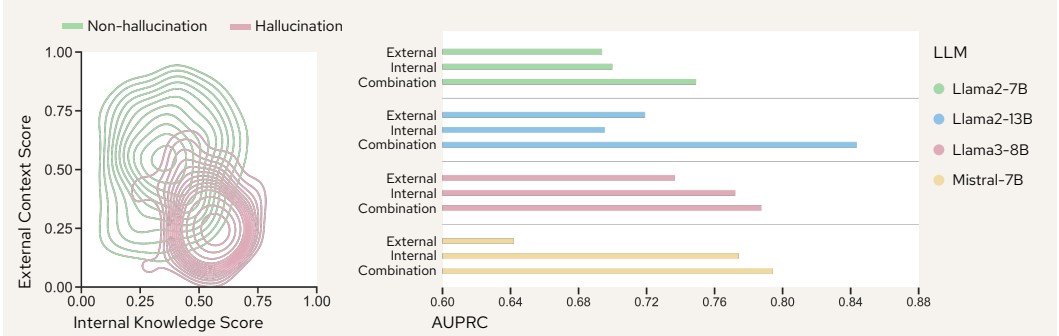

Figure 5: **Combining scores of external context and internal knowledge boosts the hallucination detection performance.** Left: 2D kernel density estimation (KDE) of the distribution of external context score and internal knowledge score of Llama2-13B responses on the RAGTruth dataset. Right: Hallucination detection performance with external/internal score only, as well as the performance of their combination.

prior attempts of quantifying external context and internal knowledge utilization, and being competitive with supervised hallucination detection models. Analyses also show that LUMINA is robust against noise in retrieved documents and can be generalized to the proxy LLM setting, demonstrating its usability in real-world scenarios.

## ETHICS STATEMENT

This work introduces LUMINA, a novel way to estimate the utilization of external context and internal knowledge when an LLM generates responses with the RAG setup. LUMINA significantly improves the performance of hallucination detection, which will help increase the reliability of RAG systems in real-world deployments and reduce the risk of sharing misinformation. Through a deeper analysis of LUMINA in the future, researchers may better understand how LLMs utilize external context and internal knowledge to generate responses. Such findings will help the community design approaches to mitigate hallucinations and create a more reliable AI system.

## REPRODUCIBILITY STATEMENT

We provide all details of the implementation of LUMINA in Appendix D, including the approximation of MMD, the selection of kernel, and the choice of random documents for measuring external context score, as well as the calibration of internal knowledge score. In Sec. 4.1, we illustrate the experimental settings, including baselines, datasets, LLMs, and evaluation metrics. The details of baselines and datasets are further provided in Appendix B and C, respectively. Furthermore, we provide the codebase of LUMINA at https://github.com/deeplearning-wisc/LUMINA. These comprehensive reports will help future studies easily reproduce our experiments.

## ACKNOWLEDGEMENT

We thank Chongdae Oh and Seongheon Park for their valuable feedback on the draft. This work was supported by Laboratory Directed Research and Development funding from Argonne National Laboratory, provided by the Office of Science, U.S. Department of Energy under Contract No. DE-AC02-06CH11357. Sharon Li acknowledges support from AFOSR Young Investigator Program under award number FA9550-23-1-0184, National Science Foundation under awards IIS-2237037 and IIS-2331669, Office of Naval Research under grant number N00014-23-1-2643, Schmidt Sciences Foundation, Open Philanthropy, and Alfred P. Sloan Fellowship. The funders had no role in study design, data collection, data analysis or interpretation, or manuscript preparation.

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

# APPENDIX

## CONTENTS

## A  BROADER IMPACTS

Beyond hallucination detection, LUMINA has broader impacts in interpretability and LLM understanding. Specifically, our proposed score validation framework in Sec. 3.3 suggests a novel way to empirically validate the finding of mechanistic interpretability, which can be used to highlight the soundness of proposed hypotheses. In addition, our proposed information processing rate in Sec. 3.2 presents a new lens for examining the internal states of LLMs. Deeper investigation of this measure could help the community better characterize how LLMs reason and leverage internal knowledge, potentially leading to more reliable training and inference processes. While our experiments focus on using LUMINA for hallucination detection, its utility extends further. For instance, it could inform the design of new training objectives or decoding algorithms aimed at mitigating hallucinations, ultimately making LLMs more reliable and trustworthy.

## B  DETAILS OF BASELINES

**(1) Token-level uncertainty:**

- **Perplexity:** This approach measured the perplexity of the generated response as uncertainty and to detect hallucinations.
- **LN-Entropy:** This approach measured sequence-level uncertainty with entropy normalized by sequence length. A higher entropy indicates greater uncertainty and a higher likelihood of hallucinations.
- **Focus:** This approach used entropy and token probability as a based score, and calibrated it by focusing only on key informative tokens and propagating the score according to the attention weight.

**(2) Cross-sample consistency:**

- **SelfCKGPT:** This approach sampled multiple responses and used an NLI model to check the logistic consistency between the target generation and additional samples. In our experiment, we follow the setting of Manakul et al. (2023) to set the sample size as 20.

- **EigenScore:** Similar to SelfCKGPT, this approach sampled multiple responses and checked the semantic consistency between the additional samples and the target generation through measuring the eigenvalues of responses' covariance matrix. In our experiment, we set the sample size as 20.

**(3) Verbalization:**

- **P(True):** This approach prompted an LLM with the generated answer and asked whether the LLM think the answer is true. The approach then estimated the probability of the "Yes" generated by the LLM.

- **RefChecker:** This approach prompted an LLM to extract claims from generation, and prompted another LLM to verify the logical consistency between each claim and reference documents. In our experiment, we use `dongyru/Mistral-7B-Claim-Extractor`, the model finetuned by Hu et al. (2024), to extract claims.

**(4) Utilization of external context and internal knowledge:**

- **ReDeEP:** For external context utilization, ReDeEP measured the cosine similarity between the generated token and topK attended tokens in retrieved documents. For internal knowledge utilization, it measured the JS divergence of the vocabulary distributions between logit lens outputs before and after FFN layers in a Transformer. At the end, it weighted summed the two scores to obtain a hallucination score.

## C   DETAILS OF DATASETS

**RAGTruth.**   The RAGTruth dataset is a human annotated hallucination detection dataset, containing 15,090 training data and 2,700 testing data. Each data point consists of a query, retrieved documents, LLM-generated answer, and span-level hallucination annotation. The dataset covers three tasks, including summarization, data to text generation, and question answering. For each query-and-documents pair, RAGTruth provides answers generated by six different LLMs, including Llama2-7B, Llama2-13B, Llama2-70B, Mistral-7B, GPT-3.5, and GPT-4. In our experiment, we also utilize the extended test set provided by Sun et al. (2025b), who curated and annotated Llama3-8B generated responses.

**HalluRAG.**   HalluRAG is an LLM annotated hallucination detection dataset for question answering. Ridder & Schilling (2025) prompted GPT-4o to generate question given sentences from Wikipedia, then used Llama2-7B, Llama2-13B, and Mistral-7B to generate answer for each question given the relevant Wikipedia article. The hallucination labels were assigned by GPT-4o with a Chain-of-Thought (CoT) prompt and verified by human. HalluRAG contains both answerable and unanswerable questions, while we only use the answerable instances for evaluation.

## D   IMPLEMENTATION DETAILS OF LUMINA

For external context utilization, we measure MMD with Eq. (6), which requires summing over the combinations of the entire vocabulary. In practice we approximate it with the top 100 tokens to reduce the computational cost. To obtain $p_{\text{ctx}'}$, in our experiment we treat the retrieved documents of another data point as the $d'$ of the target data point. In a real-world RAG system, $d'$ can be obtained by selecting random documents from the data store or retrieving less relevant documents of the query with a retrieval model.

For internal knowledge utilization, Eq. (10) computes the first information process rate of generating $a_t$ based on the next token with the highest probability. However, due to the sampling process of generation, the generated token $a_t$ is not always the highest probability token. Thus, the internal knowledge used during the generation process may not fully apply to $a_t$. To take this factor into

Table 3: **LUMINA consistently achieves a balanced precision-recall trade-off and high F1 score across datasets and LLMs.** We report the score of $\text{Prec}_{\text{Opt}}$, $\text{Recall}_{\text{Opt}}$, and $\text{F1}_{\text{Opt}}$ for LUMINA and baselines on each dataset.

| LLM | Approach | RAGTruth | | | HalluRAG | | |
| | | $\text{Prec}_{\text{Opt}}$ ↑ | $\text{Recall}_{\text{Opt}}$ ↑ | $\text{F1}_{\text{Opt}}$ ↑ | $\text{Prec}_{\text{Opt}}$ ↑ | $\text{Recall}_{\text{Opt}}$ ↑ | $\text{F1}_{\text{Opt}}$ ↑ |
|---|---|---|---|---|---|---|---|
| Llama2-7B | Perplexity | 0.5080 | 0.9867 | 0.6707 | 0.2531 | 1.0000 | 0.4040 |
| | LN-Entropy | 0.6303 | 0.7920 | 0.7020 | 0.7143 | 0.7500 | 0.7317 |
| | Focus | 0.5276 | 0.9292 | 0.6731 | 0.3077 | 1.0000 | 0.4706 |
| | SelfCKGPT | 0.5125 | 1.0000 | 0.6777 | 0.2631 | 1.0000 | 0.4167 |
| | EigenScore | 0.5201 | 0.9735 | 0.6780 | 0.4333 | 0.6500 | 0.5200 |
| | P(True) | 0.5079 | 0.9956 | 0.6726 | 0.3065 | 0.9500 | 0.4634 |
| | RefChecker | 0.5022 | 1.0000 | 0.6686 | 0.2532 | 1.0000 | 0.4040 |
| | ReDeEP | 0.6898 | 0.7478 | 0.7176 | 0.4167 | 0.7500 | 0.5357 |
| | **LUMINA** | 0.7131 | 0.7699 | 0.7404 | 0.7826 | 0.9000 | 0.8372 |
| Llama2-13B | Perplexity | 0.4926 | 0.9662 | 0.6525 | 0.1519 | 1.0000 | 0.2637 |
| | LN-Entropy | 0.6602 | 0.8164 | 0.7300 | 0.5385 | 0.5833 | 0.5600 |
| | Focus | 0.4938 | 0.9565 | 0.6513 | 0.5556 | 0.4167 | 0.4762 |
| | SelfCKGPT | 0.4801 | 0.9903 | 0.6467 | 0.3056 | 0.9167 | 0.4583 |
| | EigenScore | 0.5389 | 0.9034 | 0.6751 | 0.5833 | 0.5833 | 0.5833 |
| | P(True) | 0.6890 | 0.6957 | 0.6923 | 0.2449 | 1.0000 | 0.3934 |
| | RefChecker | 0.4600 | 1.0000 | 0.6301 | 0.2727 | 0.2500 | 0.2609 |
| | ReDeEP | 0.7772 | 0.7246 | 0.7500 | 0.4706 | 0.6667 | 0.5517 |
| | **LUMINA** | 0.7816 | 0.7778 | 0.7797 | 1.0000 | 0.7500 | 0.8571 |
| Llama3-8B | Perplexity | 0.6369 | 0.8519 | 0.7289 | - | - | - |
| | LN-Entropy | 0.5852 | 0.9465 | 0.7233 | - | - | - |
| | Focus | 0.5571 | 0.9630 | 0.7059 | - | - | - |
| | SelfCKGPT | 0.5657 | 0.9918 | 0.7205 | - | - | - |
| | EigenScore | 0.5907 | 0.9383 | 0.7250 | - | - | - |
| | P(True) | 0.5718 | 0.9342 | 0.7094 | - | - | - |
| | RefChecker | 0.5400 | 1.0000 | 0.7013 | - | - | - |
| | ReDeEP | 0.6621 | 0.7901 | 0.7205 | - | - | - |
| | **LUMINA** | 0.6988 | 0.7449 | 0.7211 | - | - | - |
| Mistral-7B | Perplexity | 0.6187 | 0.9243 | 0.7412 | 0.1702 | 0.8000 | 0.2807 |
| | LN-Entropy | 0.6890 | 0.9040 | 0.7820 | 0.8571 | 0.6000 | 0.7059 |
| | Focus | 0.7175 | 0.9004 | 0.7986 | 0.7143 | 0.5000 | 0.5882 |
| | SelfCKGPT | 0.5914 | 0.9920 | 0.7411 | 0.5385 | 0.7000 | 0.6087 |
| | EigenScore | 0.5931 | 0.9522 | 0.7309 | 1.0000 | 0.5000 | 0.6667 |
| | P(True) | 0.7030 | 0.8486 | 0.7690 | 0.3333 | 0.3000 | 0.3158 |
| | RefChecker | 0.5578 | 1.0000 | 0.7161 | 0.1266 | 1.0000 | 0.2247 |
| | ReDeEP | 0.6506 | 0.8640 | 0.7423 | 0.6250 | 0.5000 | 0.5556 |
| | **LUMINA** | 0.6600 | 0.9320 | 0.7728 | 0.9000 | 0.9000 | 0.9000 |

account, we calibrate the internal knowledge score by the ratio of probability between the generated token and the highest probability token. In the end, the calibrated internal knowledge score of $a_t$ is defined as

$$\mathcal{I}_{p_\theta}(a_t|q, d, a_{<t}) := \frac{p_\theta(a_t|q, d, a_{<t})}{p_\theta(a_{t,1}|q, d, a_{<t})} \cdot \mathcal{R}_{p_\theta}(q, d, a_{<t}). \tag{11}$$

# E ADDITIONAL EXPERIMENTAL RESULTS

## E.1 EVALUATION WITH OTHER METRICS

Table 3 shows the scores of $\text{Prec}_{\text{Opt}}$, $\text{Recall}_{\text{Opt}}$, and $\text{F1}_{\text{Opt}}$ on each dataset. The results show that LU-MINA consistently has a balanced precision-recall trade-off, where the differences between $\text{Prec}_{\text{Opt}}$ and $\text{Recall}_{\text{Opt}}$ are smaller than other baselines. Specifically, it achieves $(\text{Prec}_{\text{Opt}}, \text{Recall}_{\text{Opt}}) = (0.9, 0.9)$ on HalluRAG with Mistral-7B. This suggests that LUMINA does not over-predict hallucinations to achieve a high $\text{F1}_{\text{Opt}}$ score.

Table 4: **LUMINA achieves a competitive performance against supervised approaches.** We report the score of AUROC (ROC), Pearson's correlation coefficient (PCC), and AUPRC (PRC) for LUMINA and baselines on each dataset. The highest scores are set in **bold**.

| LLM | Approach | RAGTruth | | | HalluRAG | | |
|---|---|---|---|---|---|---|---|
| | | ROC ↑ | PCC ↑ | PRC ↑ | ROC ↑ | PCC ↑ | PRC ↑ |
| Llama2-7B | SAPLMA | 0.6508 | 0.2530 | 0.6446 | 0.8813 | **0.6710** | **0.8023** |
| | LUMINA | **0.7646** | **0.4546** | **0.7491** | **0.9153** | 0.6554 | 0.7572 |
| Llama2-13B | SAPLMA | 0.8337 | 0.5623 | **0.8466** | 0.8925 | **0.8249** | **0.8647** |
| | LUMINA | **0.8569** | **0.6041** | 0.8436 | **0.9166** | 0.6044 | 0.8497 |
| Mistral-7B | SAPLMA | **0.8073** | **0.5027** | **0.8164** | 0.9667 | **0.7920** | 0.9088 |
| | LUMINA | 0.7685 | 0.4623 | 0.7942 | **0.9899** | 0.7529 | **0.9431** |

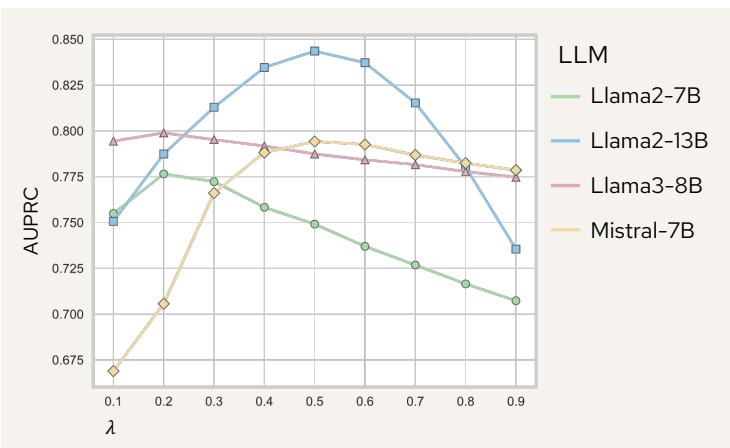

Figure 6: **A good performance of LUMINA happens with a medium $\lambda$ value**. We alter $\lambda$ in Eq. (1) to control the weight of internal knowledge score and external context score and evaluate the resulted hallucination detection performance. We conduct the experiment on the RAGTruth dataset and report the AUPRC score.

## E.2 COMPARE WITH SUPERVISED BASELINES

We further compare LUMINA with SAPLMA (Azaria & Mitchell, 2023), a supervised approach that trained a MLP model over the internal hidden states of the last generated token to classify whether the generation is hallucination or not. Following the original paper, we use hidden states at the 20th layer as input features of SAPLMA. Result in Table 4 shows that LUMINA has a competitive performance against SAPLMA and even sometimes outperforms it. Note that Table 4 doesn't show the result of Llama3-8B as the training set doesn't contain responses generated by Llama3-8B.

## E.3 PERFORMANCE WITH HYPERPARAMETER TUNING

We evaluate the hallucination detection performance with $\lambda \in \{0.1, 0.2, 0.3, 0.4, 0.5, 0.6, 0.7, 0.8, 0.9\}$. Figure 6 shows the AUPRC of different $\lambda$ on the RAGTruth dataset. The results show that the LUMINA achieves the optimal performance with varies $\lambda$ across LLMs. For Llama2-13B and Mistral-7B, setting $\lambda = 0.5$, *i.e.*, the default setting, is the optimal. While for Llama2-7B and Llama3-8B, the optimal $\lambda$ is 0.2. However, for these two models, their performance only drops less than 0.025 when setting $\lambda = 0.5$, suggesting that weighting internal knowledge and external context utilization equally is still a good practice.

Table 5: **Both components of information processing rate are important.** We report the AUROC of each component, as well as the performance of their combination.

| LLM | Layer weighting | Entropy normalization | Both |
|-----|-----------------|-----------------------|------|
| Llama2-7B | 0.7023 | 0.7164 | **0.7652** |
| Llama2-13B | 0.8007 | 0.8433 | **0.8554** |
| Llama3-8B | 0.7512 | 0.7683 | **0.7697** |
| Mistral-7B | 0.6111 | 0.6723 | **0.7679** |

### E.4 ADDITIONAL ABLATION STUDY

**Impact of MMD approximation.** When implementing LUMINA, we approximate MMD with the top $k$ tokens and set $k = 100$, aiming to balance between computational cost and approximation error. To test the impact of $k$, we ablate on $k \in \{50, 100, 500\}$ and evaluate on the RAGTruth dataset. The result shows a consistent AUROC across different $k$, with a $< 0.02\%$ difference, suggesting that LUMINA is insensitive to the choice of MMD approximation. Additionally, in cases where the computational power is limited, choosing $k = 50$ is also considerable.

**Impact of random documents.** In Section 4.3, we study the impact of noises in the retrieved and random documents. To further examine the impact of different random documents on the performance, we select 5 different random documents and use each of them to compute the hallucination score. Experiments on the RAGTruth dataset shows that the standard deviation across the 5 rounds with different random documents is $< 0.0025$, suggesting that LUMINA is very robust to the choice of random document.

**Impact of the components of the information processing rate.** Our proposed information processing rate consist of two components: layer weighting probability ratio (numerator) and entropy normalization (denominator). Table 5 shows the AUROC of ablating these two components. The result shows that both components contribute to the overall performance, justifying our design choice.

## F ERROR ANALYSIS

To analyze the failure of LUMINA, we sample 40 cases from the RAGTruth dataset that are (1) hallucinated with high-external context and low-internal knowledge scores (*i.e.*, false negative) or (2) non-hallucinated with low-external context and high-internal knowledge scores (*i.e.*, false positive). We qualitatively analyze these cases and categorize them into three groups:

**(1) Incorrect labels.** Sometimes LLMs generate fabricated content that is not sourced from the retrieved document (*e.g.*, a detailed menu of a restaurant). However, these fabricated contents are sometimes not identified by human annotators. Also, human annotators sometimes misclassify semantically equivalent content as hallucination. In these cases, the provided labels are incorrect, and LUMINA indeed correctly detects hallucination.

**(2) Generally low hallucination score for the summarization task.** We observe that many false negative samples come from the summarization task. In these cases, the LLM does generate content that contradicts the retrieved documents and has a relatively high internal knowledge score. However, since most of the generated content is still grounded in the retrieved documents, they usually have a high external score as well, resulting in a relatively low hallucination score. This observation suggests that different tasks might have different distributions of hallucination scores. A better practice is to independently evaluate the hallucination detection performance on each task.

**(3) Low quality of retrieved documents.** For the false positive cases, we observe that many of them are due to the quality issue of the retrieved documents. These documents often contain only irrelevant information or are too vague to concretely answer the query. Thus, the LLM has to reason over them and respond with "unable to answer" or use its internal knowledge to generate answers with details and examples. This results in a relatively high internal knowledge score and a low external

Table 6: **The errors of LUMINA are mainly due to incorrect labels, quality of retrieved documents, and task-dependent biases.** We report the proportion of each error type classified by GPT-5.

| Error Type | Proportion |
|---|---|
| **False Positive** | |
| Incorrect labels | 32% |
| Low quality of retrieved documents | 24% |
| Others | 44% |
| **False Negative** | |
| Incorrect labels | 16% |
| Low hallucination score for summarization task | 64% |
| Others | 20% |

Table 7: **LUMINA is more effecient than ReDeEP.** We report the average computational time (second/sample) for ReDeEP and LUMINA.

| LLM | ReDeEP | LUMINA |
|---|---|---|
| Llama2-7B | 0.86 | **0.69** |
| Llama2-17B | 1.17 | **0.88** |
| Llama3-8B | 1.13 | **0.58** |
| Mistral-7B | 0.72 | **0.54** |

context score. To address this, a future direction can focus on assessing whether the utilization of internal knowledge is necessary and correct, and using that to calibrate the hallucination score.

We extend the error analysis by sampling 50 false positive and 50 false negative cases, and prompting GPT-5 to classify the reason for error. The result in Table 6 shows that while there are edge cases that LUMINA can not handle correctly, many of the errors are due to incorrect labels and low quality of retrieved documents. For those edge cases, we observe that they usually happen when the internal knowledge and external context scores are close and when the task is more reasoning intensive. Thus, when deploying LUMINA, controlling the balance between internal knowledge score and external knowledge score according to the task might be a good practice to further increase the performance.

## G  COMPUTATIONAL RESOURCES

LUMINA is a lightweight and efficient approach, which requires only two forward passes to obtain the necessary information to compute external context and internal knowledge scores. As LUMINA does not require generating multiple samples nor training, it is easy to scale up to a large amount of data. All the experiments of LUMINA are conducted on a single Nvidia H100 GPU. The execution time of computing both external context and internal knowledge scores varies depending on the length of the response. For responses around 150 tokens, the average computational time is less than 1 second. In addition, while LUMINA requires two forward passes to compute the score, it is consistently more efficient than ReDeEP, as shown in Table 7. We believe that it is because for the external context score, ReDeEP has to store the entire attention map for every layer and use that to select the top $k$ tokens from the external context. And for the internal knowledge score, ReDeEP has to apply the logit lens before and after FFN for each transformer layer. In contrast, our external context score only requires approximating MMD at the output layer, and our internal knowledge score needs applying the logit lens only once per layer. These design choices reduce the computational cost, making LUMINA much more efficient.

