# OpenReview forum: "LUMINA: Detecting Hallucinations in RAG System with Context–Knowledge Signals"
_ICLR.cc/2026/Conference — ICLR 2026 Poster_

### Official Review · Reviewer_TLT8 · 2025-10-29

**Soundness:** 3
**Presentation:** 3
**Contribution:** 3
**Rating:** 6
**Confidence:** 4

**Summary:**

This paper proposes LUMINA, a framework for detecting hallucinations in Retrieval-Augmented Generation (RAG) systems by quantifying external context utilization and internal knowledge utilization. The authors measure external context utilization via Maximum Mean Discrepancy (MMD) between predictive distributions conditioned on retrieved vs. random documents, and internal knowledge utilization by tracking token prediction evolution across transformer layers (information processing rate). The method is validated through statistical hypothesis testing and demonstrates strong performance on RAGTruth and HalluRAG benchmarks across four open-source LLMs.

**Strengths:**

1.	Novel and principled approach: The decomposition of hallucination signals into external context and internal knowledge utilization is well-motivated. The use of MMD for context sensitivity and information processing rate for internal knowledge are creative and theoretically grounded.
2.	Statistical validation framework: The hypothesis testing approach (Section 3.3) to validate the proposed measurements is a significant contribution. This addresses a critical gap in prior work (e.g., ReDeEP) that lacked validation of whether their scores truly reflect utilization.
3.	Comprehensive experiments: Testing across 4 LLMs, 2 datasets, multiple baselines, and extensive ablations demonstrates thoroughness.

**Weaknesses:**

1.	Limited theoretical justification for Conjecture 1: The core assumption that hallucinations occur when I_pθ >> E_pθ is presented as a conjecture without formal proof. While empirically validated, stronger theoretical grounding (e.g., information-theoretic analysis) would strengthen the contribution.
2.	MMD approximation concerns: (1) The approximation using only top-100 tokens (Appendix D) may miss important distributional information; (2) No analysis of approximation error or sensitivity to this choice; (3) The choice seems arbitrary without justification.
3.	Random document selection: The paper states d' is obtained from "another data point" but doesn't discuss: (1) How this choice affects results; (2) Whether document similarity between d and d' matters; (3) Potential biases from this selection strategy
4.	Evaluation limitations: (1) No comparison with more recent methods (paper only compares with ReDeEP from 2025); (2) Missing evaluation on longer-form generation tasks; (3) HalluRAG doesn't include Llama3-8B results (unexplained gap)
5.	Computational cost: While claimed to be "lightweight" (Appendix G), requiring two forward passes with full hidden states may still be expensive for very long contexts. No comparison of computational cost vs. baselines.

**Questions:**

1.	Can you provide theoretical analysis or bounds on the MMD approximation error when using top-100 tokens?
2.	How sensitive is the method to the choice of random documents d'? Have you tried adversarial selection?
3.	Can you ablate the components of Equation (8) to show the contribution of layer weighting vs. entropy normalization?
4.	What is the computational cost comparison (wall-clock time, memory) vs. baselines like ReDeEP?
5.	Why is HalluRAG missing Llama3-8B results?
6.	Can you provide more quantitative error analysis beyond 20 sampled cases?

---

> ### Author Response · Authors · 2025-11-17
> **Author's response (Part 1)**
>
> We sincerely appreciate your feedback and insightful comments! We address the questions and comments below in detail.
>
> > `W1/Q1` Limited theoretical justification for Conjecture 1
>
> We thank the reviewer for raising this important concern. We totally agree that a theoretical justification for Conjecture 1 can strengthen the contribution. However, to our understanding, there is little theoretical finding on how the RAG system works and how LLM hallucinates; it is difficult to justify our conjecture by theory for now. In contrast, our conjecture is built on a principled intuition. That is, if a LLM requires external knowledge to complete a task and if a retriever can provide the LLM sufficient external information, the LLM should utilize those external context and ground its reasoning ability on those context. Therefore, a response in this scenario will be considered less reliable if it over-relies on the LLM's internal knowledge without a sufficient amount of external knowledge utilization. We have clarified this point in `Section 2.1`.
>
> > `W2` MMD approximation concerns
>
> Choosing the top 100 tokens aims to balance between computational cost and approximation error. To test the impact of $k$, we try $k\in\{50, 100, 500\}$ and show that **the performances remain almost consistent with a $<0.02%$ difference.** In other words, people can even choose $k=50$ to save more computational power. We have added this finding to `Appendix E.4`.
>
> > `W3/Q2` Random document selection
>
> To cast out this concern, we select 5 different random documents and compute the scores. The table below shows the average and standard deviation of the performance on the RAGTruth dataset. The result indicates that **LUMINA is very robust to the choice of random document, with only a $<0.0025$ standard deviation.** We have added this experiment to `Appendix E.4`. In our experiment, we select the random documents by using the documents of a different query. Ideally, such documents will be irrelevant and semantically different from the documents retrieved by the target query. To simulate the scenario that the random documents are somewhat similar to the retrieved document, in our experiment in `Section 4.3`, we randomly add 10%~30% sentences from the retrieved document to the random documents. The result shown in `Figure 2` suggests that LUMINA is fairly robust in such a scenario. In practice, people can use hard negative samples retrieved through a dense retriever as random documents, preventing the scenario where the retrieved and the random documents are highly similar to each other.
>
> |Model|ROC|PCC|PRC|
> |---|---|---|---|
> | Llama2-7B | 0.7624 (0.0016) | 0.4514 (0.0024) | 0.7459 (0.0021)|
> |Llama2-13B | 0.8554 (0.0016) | 0.5997 (0.0023) | 0.8405 (0.0023)|
> |Llama3-8B | 0.7440 (0.0007) | 0.4228 (0.0014) | 0.7877 (0.0012)|
> |Mistral-7B | 0.7680 (0.0006) | 0.4610 (0.0011) | 0.7941 (0.0010)|

---

> > ### Author Response · Authors · 2025-11-17
> > **Author's response (Part 2)**
> >
> > > `W4/Q5` Evaluation limitations
> >
> > To our knowledge, ReDeEP is the newest approach trying to compute the external and internal knowledge to detect hallucinations, serving as the most relevant and strong baseline. Outperforming this baseline demonstrates the effectiveness of LUMINA. We are aware that there is a more recent work, SEReDeEP [1]. However, since they did not provide code or sufficient technical details in their paper, we cannot reproduce their result. We are also aware that there are other strong baselines, such as TSV [2]. However, it is a supervised-learning-based method, which is not a fair comparison to our unsupervised approach. We are happy to include other baselines if there are specific ones in the reviewer's mind.
> >
> > For the task, we'd like to clarify that **RAGTruth is a long-form generation dataset**, with a 130 tokens average generation length. HalluRAG did not report the average length of the generation, while we observed that the answers are generally more than one sentence. We'd also like to clarify that we did not report the performance of Llama3-8B on HalluRAG in `Table 2` because **HalluRAG did not provide answers generated by Llama3-8B** (as we mentioned in the caption at `Line 325`).
> >
> > > `Q3` Can you ablate the components of Equation (8) to show the contribution of layer weighting vs. entropy normalization?
> >
> > The table below shows the AUROC of (1) layer weighting only, (2) entropy normalization only, and (3) both. The result shows that **both components contribute to the overall performance.** We have added this result to `Appendix E.4`.
> >
> > |Model|Layer weighting|Entropy normalization|Both|
> > |---|---|---|---|
> > |Llama2-7B|0.7023|0.7164|**0.7652**|
> > |Llama2-13B|0.8007|0.8433|**0.8554**|
> > |Llama3-8B|0.7512|0.7683|**0.7697**|
> > |Mistral-7B|0.6111|0.6723|**0.7679**|
> >
> > > `Q4` Computational cost
> >
> > The table below shows the computational cost (seconds per sample) of ReDeEP and LUMINA. **While LUMINA requires two forward passes, it is more efficient than ReDeEP.** We believe that it is because for the external context score, ReDeEP has to store the entire attention map for every layer and use that to select the top $k$ tokens from the external context. And for the internal knowledge score, ReDeEP has to apply the logit lens before and after FFN for each transformer layer. In contrast, our external context score only requires approximating MMD at the output layer, and our internal knowledge score needs applying the logit lens only once per layer. These design choices reduce the computational cost, making LUMINA much more efficient. We have added this result to `Appendix G`.
> >
> > |Model|ReDeEP|LUMINA|
> > |---|---|---|
> > |Llama2-7B|0.86|**0.69**|
> > |Llama2-17B|1.17|**0.88**|
> > |Llama3-8B|1.13|**0.58**|
> > |Mistral-7B|0.72|**0.54**|
> >
> >
> > > `Q6` Can you provide more quantitative error analysis beyond 20 sampled cases?
> >
> > We sample 50 false positive and 50 false negative cases and prompt GPT-5 to classify the reason for error. The tables below show the statistics of the error types. The result shows that while there are edge cases that LUMINA can not handle correctly, many of the errors are due to incorrect labels and low quality of retrieved documents, suggesting a better performance on higher-quality data. We have added this result to `Appendix F`.
> >
> > #### False Positive
> > |Error type|Proportion|
> > |---|---|
> > |Incorrect labels|32%|
> > |Low quality of retrieved documents|24%|
> > |Others|44%|
> >
> > #### False Negative
> > |Error type|Proportion|
> > |---|---|
> > |Incorrect labels|16%|
> > |Low hallucination score for summarization task|64%|
> > |Others|20%|
> >
> >
> > [1] Wang, Lei (2025). SEReDeEP: Hallucination Detection in Retrieval-Augmented Models via Semantic Entropy and Context-Parameter Fusion
> >
> > [2] Park, Seongheon et al. (2025). Steer LLM Latents for Hallucination Detection

---

### Official Review · Reviewer_VWtx · 2025-10-30

**Soundness:** 2
**Presentation:** 3
**Contribution:** 2
**Rating:** 6
**Confidence:** 3

**Summary:**

This paper introduces LUMINA, a framework for detecting hallucinations in Retrieval-Augmented Generation (RAG) systems by jointly modeling external context utilization and internal knowledge utilization. The method assumes that if an LLM effectively leverages external documents, replacing them with random ones should significantly alter the model’s output distribution. Internal knowledge usage is estimated through layerwise token evolution, measured by changes in predicted token distributions across transformer layers. LUMINA integrates these two signals into a unified hallucination score and provides a statistical validation framework. Experimental results across common RAG hallucination benchmarks demonstrate consistent improvements in AUROC and AUPRC compared to prior utilization-based detectors, with robustness under noisy or mismatched retrieval settings.

**Strengths:**

1.	Clear motivation addressing persistent hallucinations even with correct retrievals.
2.	Simple yet effective framework combining two measurable utilization signals without model retraining.
3.	Comprehensive experiments across multiple LLMs and benchmarks, with robust results under noise and model mismatch.
4.	Practical applicability — LUMINA is lightweight, reproducible, and generalizable to different RAG architectures.
5.	Statistical validation adds methodological rigor to empirical measurements, distinguishing the work from heuristic detectors.

**Weaknesses:**

1.	Assumption vulnerability — The external-context utilization metric assumes that replacing relevant documents with random ones should change generation probabilities. This fails when the model already knows the correct answer from its parametric knowledge, leading to potential false negatives.
2.	Limited generalization to reasoning tasks — The internal-knowledge utilization signal may not handle complex reasoning cases where correct responses rely on both external and internal information, potentially misclassifying reasoning-based outputs as hallucinations.
3.	Insufficient explanation for robustness — While LUMINA shows stability under noisy retrievals, the paper does not clearly justify why its metrics remain reliable when external documents are irrelevant or partially correct.
4.	Missing theoretical grounding — The framework treats token-level utilization as a direct indicator of truthfulness, but lacks analysis connecting these signals to semantic correctness or factual alignment.
5.	Incomplete discussion of failure cases — The paper could better analyze when and why LUMINA misclassifies or underperforms, especially under low retrieval relevance.

**Questions:**

1.	How does LUMINA distinguish between cases where the model truly ignores external context and cases where external and internal knowledge simply overlap?
2.	Could the authors provide examples where correct answers yield low context-utilization scores, and explain how such cases are handled?
3.	How might the proposed framework adapt to reasoning tasks that integrate both context and internal knowledge, such as multi-hop question answering?
4.	Why does LUMINA remain robust under noisy retrievals, given that both proposed metrics are highly dependent on the correctness of external documents?
5.	Would integrating interpretability analyses (e.g., attention visualization or neuron probing) strengthen the causal link between the measured utilization signals and factual reliability?

---

> ### Author Response · Authors · 2025-11-17
> **Author's response (Part 1)**
>
> We sincerely appreciate your feedback and insightful comments! We address the questions and comments below in detail.
>
> > `W1` Assumption vulnerability
>
> We thank the reviewer for raising this concern. To clarify, **our paper focuses on the scenarios where the external context is essential for completing the task**, such as news summarization, information extraction given a json file, and question answering, where the required information emerged after the model's release date. In these tasks, RAG is necessary as a model can not rely on their internal knowledge alone. For tasks that are relatively trivial and can be solved by a model on its own parametric knowledge, we believe that RAG may not be necessary, and thus, it is not what LUMINA is designed for. We have clarified this in `Section 2.1`.
>
> > `W2` Limited generalization to reasoning tasks
>
> Our hallucination score does not assume that there shouldn't be any utilization of internal knowledge, and we are aware that in reasoning tasks, the internal knowledge score might be higher. However, we believe that in such cases, the reasoning should be grounded in the external context, resulting in both high external and internal knowledge scores. We'd also like to highlight that RAGTruth contains a task of data-to-text, which requires the model to reason according to the provided json file. And LUMINA has a generally good performance on that task compared to other tasks like summarization and question answering. In addition, in `Figure 4`, the distribution of external and internal knowledge scores suggests that a generation is marked as hallucination by LUMINA mainly if it has a disproportionally high internal knowledge score while with a low external score, demonstrating its potential for generalizing to reasoning-intensive tasks. We have added the discussion of the result to `Section 4.4`.
>
> > `W3/Q4` Why does LUMINA remain robust under noisy retrievals, given that both proposed metrics are highly dependent on the correctness of external documents?
>
> We'd like to clarify that in our noise injection setting, injecting noise does not change the correctness of the retrieved documents, nor does it make the retrieved documents totally irrelevant. Instead, it only makes the retrieved documents less sufficient and makes the random documents semantically slightly close to the retrieved documents. We use this setting to demonstrate that the assumption of "sufficient retrieved documents" is not necessary for LUMINA, making it flexible and robust in real-world scenarios. However, we believe that the retrieved documents can not be completely irrelevant or incorrect, which violates the basic assumption of RAG, and thus, LUMINA is not guaranteed to work. Based on this premise, LUMINA is robust because a LLM should not completely rely on its internal knowledge when the external context is insufficient. Instead, its internal knowledge or reasoning capacity should be used to fill in the gap of the insufficiency of the external context. Thus, although the external context score will slightly decrease and the internal knowledge score will slightly increase under the noisy setting, we can still identify hallucinations when the internal knowledge score significantly outweighs the external context score.
>
> > `W4` Theoretical grounding
>
> We thank the reviewer for raising this important concern. We totally agree that a theoretical grounding can strengthen the contribution. However, to our understanding, there is little theoretical finding on how the RAG system works and how LLM hallucinates; it is difficult to establish the full formal link by theory for now. Instead, LUMINA operates under a more modest and empirically validated conjecture: if a LLM requires external knowledge to complete a task and if a retriever can provide the LLM sufficient external information, the LLM should utilize those external context and ground its reasoning ability on those context. We verify the soundness of the approach in `Section 3.3` through rigorous statistical hypothesis testing.

---

> > ### Author Response · Authors · 2025-11-17
> > **Author's response (Part 2)**
> >
> > > `Q5` Would integrating interpretability analyses (e.g., attention visualization or neuron probing) strengthen the causal link between the measured utilization signals and factual reliability?
> >
> > We appreciate this insightful suggestion. We agree that interpretability tools could potentially provide complementary evidence about how models route information between internal knowledge and retrieved context.
> >
> > To examine whether our external-utilization metric aligns with established interpretability signals, we conducted an additional analysis comparing the external-context score with the model’s attention over external-context tokens. The table below reports the token-level Spearman correlation between these two signals.
> >
> > While the correlations are positive across all models, they are relatively modest. This is expected and aligns with prior findings that attention is an imperfect and noisy proxy for actual information use [1]. Attention distributions reflect where a model looks, not necessarily which information it relies on for generation. In contrast, LUMINA’s utilization metric captures a different—and in many ways more direct—dimension: how strongly the model’s output behavior depends on the retrieved evidence. This reinforces its value as a complementary signal.
> >
> > |LLM|Spearman ρ|p-value|
> > |---|---|---|
> > |Llama-2-7B|0.109|<0.005|
> > |Llama-2-13B|0.148|<0.005|
> > |Llama-3-8B|0.175|<0.005|
> > |Mistral-7B|0.212|<0.005|
> >
> > > `W5` Incomplete discussion of failure cases
> > > `Q2` Could the authors provide examples where correct answers yield low context-utilization scores, and explain how such cases are handled?
> >
> > We thank the reviewer for asking this important question, and we'd like to highlight that **we already included error analysis in Appendix F**. Specifically, we identify three main reasons for when and why LUMINA misclassifies. Among them, the main reason that LUMINA yields a low external context score and a high internal knowledge score is that the retrieved documents are irrelevant or are too vague. In such cases, the LLM has to reason over them and respond with "unable to answer" or use its internal knowledge to generate answers. While we also observe some edge cases that make a correct answer have an internal knowledge score higher than the external context score, they usually happen when the two scores are close and when the task is more reasoning-intensive. Thus, when deploying LUMINA, controlling the balance between internal knowledge score and external knowledge score according to the task might be a good practice to further increase the performance. We have added this discussion to `Appendix F`.
> >
> >
> > [1]: Serranom, Sofia and Smith, Noah A. (2019). Is Attention Interpretable?

---

> ### Comment · Reviewer_VWtx · 2025-11-28
>
> I thank the authors for their comprehensive responses. I would like to keep the original score.

---

> > ### Author Response · Authors · 2025-11-28
> >
> > Dear Reviewer VWtx,
> >
> > Thank you again for your positive feedback. We are happy to address your concerns. If you have further comments, we're glad to discuss them with you.
> >
> > Best regard,
> >
> > Authors

---

### Official Review · Reviewer_VZPC · 2025-10-31

**Soundness:** 3
**Presentation:** 4
**Contribution:** 3
**Rating:** 6
**Confidence:** 4

**Summary:**

This paper propose LUMINA, a framework that detects hallucinations in RAG systems with external and internal context
utilization metrics. The external context utilization metric measure the maximum mean discrepancy between two next token probability distributions conditioned on retrieved documents and random documents. The internal knowledge utilization metric introduce the idea of information processing rate by looking at the ratio of the most probable output token’s probability across transformer layers and
use it to determine the amount of utilized internal knowledge when generating the next token. The experiments show that LUMINA outperforms both score-based and learning based methods in hallucination detection.

**Strengths:**

1. This paper attributes the hallucination problem in RAG systems to LLMs' preference for utilizing internal knowledge versus external retrieval information. It argues that a good RAG system should pay more attention to external retrieval information, and designs two related metrics to quantify the ability to utilize external information and the ability to utilize internal knowledge. The idea is interesting and the metric design logic is sound.

2. This paper presents a framework to statistically validate the metrics, showing the soundness of the evaluation approach.

3. The experimental results show that the proposed evaluation approach works well on relevant benchmarks.

**Weaknesses:**

1. Although Section 3.2 describes the relationship between internal knowledge utilization and output token probability across transformer layers, it is not clear how the internal knowledge works in black-box LLMs. Could you provide some concrete examples demonstrating that the internal knowledge utilization metric effectively captures the overuse of the internal knowledge?

2. The experiments are conducted on outdated models, with the most recent being Llama 3, which was released 1.5 years ago. Could you provide additional experimental results on newer models such as Qwen 3 or Llama 3.2 to demonstrate the consistent superiority of your approach?

3. How about the evaluation resource cost of LUMINA? There should be a comparison between LUMINA and other related works.

**Questions:**

1. If a model can use internal knowledge to solve many RAG queries, how should we rate this model under LUMINA?

---

> ### Author Response · Authors · 2025-11-17
> **Author's response (Part 1)**
>
> We sincerely appreciate your feedback and insightful comments! We address the questions and comments below in detail.
>
> > `W1` It is not clear how the internal knowledge works in black-box LLMs. Could you provide some concrete examples demonstrating that the internal knowledge utilization metric effectively captures the overuse of the internal knowledge?
>
> Thank you for raising this point. We agree that internal knowledge in black-box LLMs is not directly observable. LUMINA’s contribution is precisely to provide measurable signals of when the model is relying more heavily on its internal knowledge than on the retrieved context.
>
> To make this clearer, we provide concrete examples illustrating how the internal-knowledge utilization metric captures such overuse. Briefly, these patterns arise consistently across tasks:
>
> #### Case 1: Ambiguous QA
>
> ```
> Question: temperature bucharest
> Documents:
>
> passage 1:Today’s and tonight’s Bucharest, Romania weather forecast, weather conditions and Doppler radar from The Weather Channel and Weather.com [...]
>
> passage 2:CLIMATE: AVERAGE MONTHLY WEATHER IN Bucharest, Romania. Bucharest has a humid continental climate. Winters can get windy and temperatures often dip below 0 °Celsius (32 °Fahrenheit), sometimes even dropping to minus 20 °C (4 °F). [...]
>
> LLM's response: Based on the provided passages, I can answer the following question:\n\nQuestion: What is the current temperature in Bucharest?\n\nPassage 1 provides the current weather conditions for Bucharest, including the temperature:\n\n\"Today's temperature in Bucharest is 22°C (72°F), humidity is 82%, wind speed is 6 mph (9 km/h).\"\n\nTherefore, the current temperature in Bucharest is 22°C (72°F).
>
> External context score: 0.1109
> Internal knowledge score: 0.3676
> ```
>
> In this case, the question is ambiguous, and the documents do not contain information to infer the current weather in Bucharest. Here, the LLM relied on its model priors to generate a response without grounding in the external context.
>
>
>
> #### Case 2: Information extraction
>
> ```
> Question: Write an objective overview about the following local business based only on the provided structured data in the JSON
> Documents:
>
> {
>   "name": "The Good Bar",
>   "address": "5650 Calle Real",
>   "city": "Goleta",
>   "state": "CA",
>   "categories": "Bars, Nightlife",
>   "attributes": {
>     "OutdoorSeating": true,
>     "Music": false,
>     "Ambience": {
>       "divey": null,
>       "intimate": null,
>       "trendy": null,
>       "upscale": false,
>       "classy": true,
>       "casual": true
>     }
>   },
>   "business_stars": 4.0,
>   "review_info": [
>     {
>       "review_stars": 4.0,
>       "review_text": "A nice full bar located inside of the Kimpton Hotel. [...] They are fully stocked with all of the yummy basic liquors to make your cocktail of choice. [...]"
>     },
>     {
>       "review_stars": 4.0,
>       "review_text": "In the ultimate suburbia known as Goleta, Good Bar isn't a bad spot to grab a cocktail. [...] I enjoy the eclectic cocktail selection on their menu. [...]"
>     },
>     [...]
>   ]
> }
>
> LLM's response: The Good Bar is a popular bar located inside the Kimpton Hotel in Goleta, California. It offers a wide range of cocktails, craft beers, and wines, with an emphasis on quality and variety. [...] Based on the reviews, The Good Bar is a great spot for those looking for a trendy and upscale bar experience with a wide selection of drinks and a lively atmosphere. [...]
>
> External context score: 0.1664
> Internal knowledge score: 0.4939
> ```
>
> In this case, the LLM mentioned craft beers and wines, stating that the bar is both trendy and upscale. While these are common words associated with the descriptions and comments of a bar, they were not provided by the document, indicating that the LLM relied on its model priors rather than the external context.
>
>
> #### Case 3: News summarization
>
> ```
> Question: Summarize the following news
> Document:
> They come from more than 20 countries, drawn to Libya as the funnel to Europe. [...]  Motivations among the tens of thousands making the trek to the Mediterranean coast are as many and varied as the nationalities involved, according to researchers and human rights groups. [...]
>
> LLM's response: Thousands of migrants from Africa and the Middle East are attempting to reach Europe by crossing the Mediterranean Sea, driven by conflicts, economic hardship, and the perception of better opportunities. [...] The EU has recently implemented measures to address the issue, including increasing funding for search and rescue operations and supporting the Libyan coast guard. [...]
>
> External context score: 0.2673
> Internal knowledge score: 0.5846
> ```
>
> In this case, the LLM mentioned, "The EU has recently implemented measures to address the issue [...]," which does not exist in the article. Although the statement might be true and exist in the LLM's training data, mentioning it in the summarization is a hallucination.

---

> > ### Author Response · Authors · 2025-11-17
> > **Author's response (Part 2)**
> >
> > > `W2` Additional experimental results on newer models
> >
> > Our main experiment was conducted on older models (e.g., Llama2-7B, Llama2-13B, Llama3-8B, and Mistral-7B) because the **annotated answers provided in RAGTruth and HalluRAG were originally generated by these older models**. Following the common evaluation practice in uncertainty-based hallucination detection to ensure a fair comparison, we use only a model to calculate the score **for the answers generated by the same model.**
> >
> > To show the generalizability, we further test LUMINA with **Qwen3-4B** and **Llama4-17B** on RAGTruth and report the AUROC scores. Specifically, following the cross-model setting in `Figure 3`, we separately report the scores for responses generated by different LLMs. The results shown in the table below indicate that the **newer models also perform well across settings.** We have updated the new results on `Figure 3` and discussed the result in `Line 450-460`.
> >
> > |Detection LLM\Generation LLM|Llama2-7B|Llama2-13B|Llama2-70B|Llama3-8B|Mistral-7B|GPT-3.5|GPT-4|
> > |---|---|---|---|---|---|---|---|
> > |Qwen3-4B|0.7422|0.8192|0.7654|0.6949|0.7895|0.7668|0.7778|
> > |Llama4-17B|0.7091|0.8552|0.7614|0.7202|0.7421|0.7746|0.8050|
> >
> > > `W3` Computational cost
> >
> > The table below shows the computational cost (seconds per sample) of ReDeEP and LUMINA. **While LUMINA requires two forward passes, it is more efficient than ReDeEP.** We believe that it is because for the external context score, ReDeEP has to store the entire attention map for every layer and use that to select the top $k$ tokens from the external context. And for the internal knowledge score, ReDeEP has to apply the logit lens before and after FFN for each transformer layer. In contrast, our external context score only requires approximating MMD at the output layer, and our internal knowledge score needs applying the logit lens only once per layer. These design choices reduce the computational cost, making LUMINA much more efficient. We have added this result to `Appendix G`.
> >
> > |Model|ReDeEP|LUMINA|
> > |---|---|---|
> > |Llama2-7B|0.86|**0.69**|
> > |Llama2-17B|1.17|**0.88**|
> > |Llama3-8B|1.13|**0.58**|
> > |Mistral-7B|0.72|**0.54**|
> >
> >
> > > `Q1` If a model can use internal knowledge to solve many RAG queries, how should we rate this model under LUMINA?
> >
> > We'd like to emphasize that in our paper, **we focus on the scenarios where the external knowledge is essential for completing the task**, such as news summarization, information extraction given a json file, and question answering, which requires information emerging after the model’s release date. In these settings, RAG is necessary because the model cannot rely solely on its internal knowledge. For tasks that are relatively trivial and can be solved by a model on its own, we believe that RAG may not be necessary, and therefore, such cases fall outside the scope of what LUMINA is designed for. We have clarified this point in the paper. We have clarified this in `Section 2.1`.

---

> > > ### Comment · Reviewer_VZPC · 2025-11-26
> > >
> > > Thank you to the authors for the response. My concerns and questions are addressed.

---

> > > > ### Author Response · Authors · 2025-11-28
> > > >
> > > > Dear Reviewer VZPC,
> > > >
> > > > Thank you again for your positive feedback. We are happy to address your concerns.
> > > >
> > > > Best regard,
> > > > Authors

---

### Author Response · Authors · 2025-11-17
**General response**

Dear Reviewers, AC, SAC, and PC,

We extend our sincere gratitude for the time and effort you have dedicated to reviewing our manuscript.

In this paper, we propose a new approach, LUMINA, to detect hallucinations in RAG systems. We are grateful for reviewers' recognition of the novelty and the soundness of the approach, as well as the comprehensiveness and the rigor of our experiments and validation. For convenience, we summarize below the key strengths, along with our changes in the manuscript in response to the concerns:

---

#### Key strengths noted by the reviewers:

- **S1:** **Clear motivation and problem framing**. (R1 and R2)
- **S2:** **Novel, principled, and logically sound metrics** for external context utilization and internal knowledge utilization (All reviewers)
- **S3:** **Comprehensive experiments** across multiple LLMs and benchmarks. (All reviewers)
- **S4:** The statistical validation framework adds **methodological rigor** to empirical measurements. (All reviewers)
- **S5:** **Practical applicability -- lightweight, reproducible, and generalizable** to different RAG architectures. (R2)

---

#### Key revisions in response to the comments:


We have revised the paper accordingly, and the major changes are summarized below:

- **[R1]** We update `Figure 3` to incorporate the result of Qwen3-4B and Llama4-17B.
- **[R1, R2, R3]** We clarify the problem setting and Conjecture 1 in `Section 2.1`.
- **[R1, R3]** We report the comparison of computational cost between ReDeEP and LUMINA in `Appendix G`.
- **[R2]** We discuss the generalization to reasoning tasks in `Section 2.1` and `Section 4.4`.
- **[R2, R3]** We provide further qualitative and quantitative error analysis in `Appendix F`.
- **[R3]** We report additional ablations in `Appendix E.4`, including the MMD approximation, random documents selection, and the components of the information processing rate.

All the added content has been marked in red.
(* We refer to Reviewer VZPC as R1, Reviewer VWtx as R2, and Reviewer TLT8 as R3.)

---

### Meta-Review · Area_Chair_MboT · 2025-12-05

**Summary:**

This paper propose LUMINA, a framework that detects hallucinations in RAG systems with external and internal context utilization metrics.  The method is accompanied by a statistical validation framework and is evaluated across four open-source LLMs and two benchmarks. All reviewers found the motivation clear and the methodology novel.

Weaknesses raised include:

* need for clearer concrete examples of internal-knowledge overuse, more experiments on newer models, and evaluation cost comparison.
* vulnerability of assumptions, limited discussion of reasoning tasks, incomplete explanation of robustness under noise, lack of deeper failures analysis.
* limited theoretical justification of Conjecture 1, concerns about MMD approximation, random document selection strategy, missing recent baselines, insufficient quantitative error analysis.

However, after rebuttal, nearly all concerns were addressed with concrete new analyses, added experiments, or clarifications. Importantly, Reviewer VZPC and Reviewer VWtx  kept a positive borderline score in discussion period.

Overall, this paper  presents a useful framework for hallucination detection, and the rebuttal successfully resolved the main technical concerns. So, I recommend to accept this paper.

**Reviewer Concerns:**

* The authors provided three detailed, concrete examples where internal knowledge dominates—even quoting internal/external scores.
Added new results for Qwen3-4B and Llama4-17B, also a full computational cost table.

* The authors clarified scope: LUMINA targets tasks where external context is required, consistent with RAGTruth and HalluRAG.

**Reviewer Scores:**

Given the rebuttal outcomes and follow-up discussion:
* Reviewer VZPC: Their concerns were fully resolved; they explicitly approved the responses. Score would remain 6.
* Reviewer VWtx: Appreciated the comprehensive response; while kept the score, no further negative concerns. Score likely remains 6.
* Reviewer TLT8: All major concerns addressed; theoretical objection acknowledged but not fatal. Score remains 6, supporting acceptance.

---

### Decision · Program_Chairs · 2026-01-26

Accept (Poster)